# BNS: Building Network Structures Dynamically for Continual Learning

**Qi Qin[1,3], Han Peng[3], Wenpeng Hu[2], Dongyan Zhao[3,*] and Bing Liu[4,*]**

[1] Center for Data Science, AAIS, Peking University
[2] Department of Information Science, School of Mathematical Sciences, Peking University
[3] Wangxuan Institute of Computer Technology, Peking University
[4] Department of Computer Science, University of Illinois at Chicago
{qinqi, phan, wenpeng.hu, zhaody}@pku.edu.cn, liub@uic.edu

## Abstract

Continual learning (CL) of a sequence of tasks is often accompanied with the *catastrophic forgetting* (CF) problem. Existing research has achieved remarkable results in overcoming CF, especially for *task continual learning*. However, limited work has been done to achieve another important goal of CL, *knowledge transfer*. In this paper, we propose a technique (called BNS) to do both. The novelty of BNS is that it dynamically builds a network to learn each new task to overcome CF and to transfer knowledge across tasks at the same time. Experimental results show that when the tasks are different (with little shared knowledge), BNS can already outperform the state-of-the-art baselines. When the tasks are similar and have shared knowledge, BNS outperforms the baselines substantially by a large margin due to its knowledge transfer capability.

## 1   Introduction

Continual learning (CL) incrementally learns a sequence of tasks in a neural network. Each task consists of a set of classes to be learned together. CL is often accompanied by the *catastrophic forgetting* (CF) problem [38]. Two main settings for CL have been extensively studied, namely, *task continual learning* (Task-CL) and *class continual learning* (Class-CL). Task-CL learns a classifier for each task. In testing, the task id is provided for the test data so that the specific task model can be applied to assign a class to the test sample. Although Class-CL also learns a sequence of tasks, in testing the task id is not provided for each sample. Note that there is also a less known or studied CL setting called *domain continual learning* (Domain-CL), in which all tasks (or domains) share the same set of classes [22] and no task id is provided in testing. This paper focuses on Task-CL.

A large amount of research has been done on overcoming CF [8]. The main existing techniques include the regularization-based methods [63, 11, 25, 18], replay-based methods [43, 18, 56] and structure-based methods [50, 61, 39]. Regularization-based methods are primarily designed to evaluate the importance of the parameters learned from old tasks, and then use some mechanisms to protect those important parameters so that the performance of old tasks will not be affected much in the process of learning new tasks. Replay-based methods use part of the data cached in the past or train a generator to generate some similar past data to help maintain the performance of the old tasks in learning new tasks. Structure-based methods allow the neural network to adaptively adjust the network structure during the learning process or to add a learning module that does not share in

---

*Corresponding authors. The main part of the work was done when Bing Liu was at Peking University on leave of absence from University of Illinois at Chicago.

35th Conference on Neural Information Processing Systems (NeurIPS 2021).

the continual learning process. In the Task-CL setting, several techniques have overcome CF, e.g, HAT [50] (see Section 4.3).

However, another major objective of CL, *knowledge transfer across tasks* [8, 21, 47], has received relatively little attention. An ideal Task-CL algorithm must do well on both (preventing CF and transferring knowledge). Being able to prevent CF alone is far from satisfactory. We will see in Section 4.3 that the knowledge transfer ability of the current Task-CL algorithms is very weak. This paper proposes a technique based on reinforcement learning (RL) called BNS (Building Network Structures dynamically for CL) to explicitly perform both CF prevention and knowledge transfer at the same time and automatically. When the tasks are different with little shared knowledge, BNS should prevent CF. When the tasks are similar with shared knowledge, BNS should transfer knowledge across tasks to achieve higher accuracy than without knowledge transfer.

BNS has three components: *a neural structure search agent*, *a set of actions*, and *the environment*. The environment includes *the current task data*, *a continual learner*, *a knowledge repository* and *a replay buffer*. We propose to use the similarity of the new task and the old tasks as the state representation in the environment. This enables knowledge transfer as more similar tasks tend to have more shared knowledge to be transferred across tasks. For each new task, the agent uses the state to sample a sequence of actions to construct a new continual learner for the task, which includes building a new network structure and selecting the past knowledge (*i.e.,* parameters) from the knowledge repository to initialize the parameters. The continual learner then learns the current task. After learning, a specially designed reward is computed based on the current task validation data and the saved data of old tasks in the replay buffer (to avoid CF and to transfer knowledge to the current task). The reward is used to guide the training of the agent through RL. Thus, BNS achieves both CF avoidance and knowledge transfer at the same time. The final model learned by the continual learner not only performs the current new task well but also maintains the performance of the old tasks.

Experimental results on five datasets MNIST, CIFAR10, CIFAR-100, F-EMNIST and F-Celeba show that BNS markedly outperforms the existing state-of-the-art Task-CL baselines. For datasets (F-EMNIST and F-Celeba) with high task similarities, BNS can leverage knowledge transfer to improve the accuracy substantially compared to the baselines. Even for datasets (MNIST, CIFAR10, and CIFAR-100) with very different/dissimilar tasks, apart from overcoming CF, it can also improve the accuracy in most cases, which existing Task-CL baselines have difficulty to do.

## 2 Related work

Existing CL approaches mainly focus on overcoming CF. Limited work has been done to explicitly leverage the knowledge gained in the continual learning process. In general, these methods can be divided into three categories: regularization-based, memory-based, and structure-based methods.

*Regularization-based methods* deal with CF by keeping the important parameters for old tasks minimally modified in the new task learning. For example, EWC [25] uses fisher information to quantify the importance of parameters to old tasks, and selectively alters the learning rates of parameters to protect the parameters. Many papers dealing with CF employ similar ideas [63, 13, 3, 44, 60, 23, 40, 11, 1, 48]. In the extreme, [62] finds orthogonal projections of weight updates that do not disturb the weights of old tasks. Many techniques also use knowledge distillation [33, 59, 6, 4, 35, 29, 53], which is another family of methods that relies on regularization, i.e., knowledge distillation loss.

*Memory-based methods* save a small number of training examples of each old task or generate some pseudo-examples of the old tasks to be used to jointly train with the current task data [56]) This approach is also called *experience replay*. In the data memorization paradigm, the representative work includes GEM [36], A-GEM [7], and many others [46, 43, 59, 45, 10, 17]. In the data generation paradigm, there is also an extensive research [51, 49, 20, 58, 18, 31, 39, 15], which typically learns a generator to generate pseudo old data. Our work also saves some training examples, but they are used in reward calculation in our setting rather than in training as in the existing methods.

*Structure-based methods* dynamically expand the network in learning each new task [33, 61, 60, 32, 39] or use masks over parameters to activate a subset of the network [50, 21, 37], which BNS also does. In particular, MNTDP [55] learns each task by expanding the existing network resulted from the previous tasks by adding modules and selectively connect them to the existing network. DEN [61]

is similar to MNTDP, although the specific details are different. Both these approaches are not based on reinforcement learning (RL). Our BNS is very different as BNS is based on RL and it constructs a new network (continual learner) for the new task and initializes it based on 5 actions, not by adding modules to the existing network. RCL [60] is an RL-based method. However, its state is a fixed empty embedding. RCL has only one action, which determines how many filters should be added in each layer for the new task. Its reward is only the new task validation data accuracy and the model complexity. Our BNS uses task similarity as the state of the environment and has 5 actions. BNS's reward considers both forgetting and transfer and thus can simultaneously tackles both catastrophic forgetting and knowledge transfer. Recently, pre-trained feature extractors have also been employed to improve the CL accuracy [19, 22]. BNS does not use any pre-trained feature extractor.

Earlier works in lifelong learning focused mainly on knowledge transfer [54, 47, 8]. However, since they use traditional learning methods such as regression [47] and naive Bayes [9, 57] to build an independent model for each task, there is no CF. In deep learning, the recent system CAT [21] learns a sequence of mixed similar and dissimilar tasks and does knowledge transfer among detected similar tasks. However, it does not do knowledge transfer among dissimilar tasks. BNS can transfer knowledge regardless of task similarity. Progressive Network [46] tries to perform forward transfer. It builds a model for each task and then connects all the models. However, it cannot do backward transfer and its network size grows quadratically in the number of tasks. It is not scalable. Several continual sentiment classification methods have focused on knowledge transfer as sentiment tasks are similar [42, 22]. However, they do not deal with dissimilar tasks for which CF is a major issue.

Our method is also related to neural architecture search (NAS) as we search for an optimal network structure. The goal of traditional NAS is to replace the manually designed models to find an advanced network structure for a specific task. A NAS method usually employs a reinforcement learning (RL) method [64, 52] to select actions to construct the final neural architecture. However, this method usually requires considerable computing power. To solve the problem of high computational cost, a weight sharing strategy has been proposed [41, 34, 5, 14]. ENAS [41] is a weight sharing algorithm using RL. After training, the parameters of the previously sampled network structure are retained. When the newly sampled network contains some substructure of the previous network, the new network will reuse some parameters of the previous network substructure. Our method borrows the idea of ENAS, which has not been used in CL before. The major difference between our work and ENAS is that our policy exploits task similarities to enable our method to transfer knowledge across tasks as similar tasks tend to have shared knowledge that can be transferred to enable new tasks to learn better. Further, our actions and reward computation are specifically designed to enable BNS to achieve both knowledge transfer and CF avoidance at the same time, which ENAS does not do.

## 3   Method

Our Task-CL setting is as follows. We incrementally learn a sequence of $N$ tasks. Each task $t$ has a training dataset $D^t_{train} = \{(x^t_j, y^t_j)\}^{n_t}_{j=1}$, where $x^t_j$ is an input instance and $y^t_j$ is its class label, and $n_t$ is the number of training examples of the $t$-th task. Similarly, the test dataset and validation dataset of task $t$ are denoted by $D^t_{test}$ and $D^t_{valid}$ (the use of the validation data will become clear shortly). We denote the datasets of task $t$ by $D^t = \{D^t_{train}, D^t_{test}, D^t_{valid}\}$. Our goal is to design a Task-CL algorithm that can adaptively determine the network structure for each new task $t$ to selectively transfer the previously learned knowledge to overcome CF and to learn the new task $t$ better. We use reinforcement learning to help achieve our goal.

**Overview of BNS:** As mentioned in the introduction, the proposed reinforcement learning based Task-CL algorithm BNS consists of three components: *a neural structure search agent*, *a set of actions*, and *an environment*. When learning the new task $t$, the environment consists of the *current task dataset* $D^t$, *a continual learner* $f(\cdot; \theta_t)$, *a knowledge repository* $\mathcal{K}$ and *a replay buffer* $\mathcal{T}$. At the high level, BNS works as follows. **First**, the current task dataset $D^t$ and the cached data of each old task in the replay buffer $\mathcal{T}$ are used to compute their similarity. The set of similarities of the new task with all old tasks forms the current environment state representation $S_t$. **Second**, the agent takes $S_t$ as input to sample an action sequence $a_t$ to build a new continual learner $f(\cdot; \theta_t)$. The actions determine the network structure and parameter initialization of the continual learner. **Third**, the new continual learner is trained using the training set $D^t_{train}$ in $D^t$, and then calculates the reward $r^t$ for the model using the validation set $D^t_{valid}$ of $D^t$ and the saved old task data in the replay buffer $\mathcal{T}$. **Fourth**, it uses the reward to update the agent via reinforcement learning.

The above four steps are repeated to train the agent $I$ iterations or times on task $t$ until it converges. After the agent is well trained, we select the continual learner with the highest reward as the final model for task $t$. This model not only performs well on the current task but also maintains the performance of the old tasks. After learning task $t$, the validation dataset $D_{valid}^t$ and the parameters of $f(\cdot; \theta_t)$ are saved in the replay buffer $\mathcal{T}$ and the knowledge repository $\mathcal{K}$ respectively. Note that at any time, we only keep the trained model of the current continual learner for the last task.

### 3.1 Environment, Agent, Action, State and Reward

**Environment:** It contains the current task dataset $D^t$, the current continual learner $f(\cdot; \theta_t)$, the replay buffer $\mathcal{T}$ and the knowledge repository $\mathcal{K}$. BNS uses a simple network topology, $f(\cdot; \theta_t)$, i.e., a network with $L$ shared layers (shared among all the learned tasks) and a task specific layer (*i.e.*, the last classification layer) for each task. In learning the current task $t$, we calculate the similarity $s_{t,i}$ between task $t$ and each old task $i$ to obtain $S_t = [s_{t,0}, ..., s_{t,t-1}]$. We regard $S_t$ as the current state representation of the environment. Specifically, we use a pre-trained feature extractor[2] $F(\cdot)$ to encode each instance of the current task's validation set $D_{valid}^t$ and the saved data of previous tasks in the replay buffer $\mathcal{T}$. We then compute the current data representation $X_t$ using the validation set $D_{valid}^t$ of task $t$ and the data representation $X_k$ of the saved validation set $D_{valid}^k \in \mathcal{T}$ of the old task $k$ in the replay buffer $\mathcal{T}$. After that, we compute the similarity $s_{t,k}$ of $X_t$ and $X_k$.

$$X_t = \frac{1}{m_t} \sum_{j=1}^{m_t} F(x_{t,j}), \quad X_k = \frac{1}{m_k} \sum_{j=1}^{m_k} F(x_{k,j}), \quad s_{t,k} = \text{Similarity}(X_t, X_k). \tag{1}$$

where $m_k$, $m_t$ are the number of validation examples and Similarity can be any similarity function such as cosine, KL divergence or $L_p$ norm. When using KL divergence, we use the softmax function to convert $X_t$ and $X_k$ into probability distributions. This paper uses the three functions together through concatenation as the measure of similarity.

**Agent:** The agent aims to produce a sequence of actions to build an optimal network for the continual learner $f(\cdot; \theta_t)$ for the current task $t$. In each training iteration of the agent for task $t$, it samples an action sequence to construct each layer of the network of $f(\cdot; \theta_t)$ using the current environment state $S_t$. Since there is correlation between the adjacent layers in the neural network, the actions for building the neural network should have a front-to-back dependency. Thus, BNS uses a LSTM [16] based network as the agent to decide the action for each layer of the current continual learner $f(\cdot; \theta_t)$. The agent is trained with policy gradient by maximizing the expected reward computed using the current task performance and the CF avoidance on the old task data in the replay buffer $\mathcal{T}$.

**Action:** BNS has two types of actions: *layer-wise actions* (*i.e.,* "reuse", "adaptation", "new", "fuse") and *element-wise action* (*i.e.,* "mask"). They decide how each layer of the continual learner should be constructed by leveraging the shared layers' weights of each old task learned by its continual learner stored in $\mathcal{K}$. For the layer-wise actions: "**reuse**" makes the new task use the same parameters as a previous task (sampled over the corresponding layers of all the old tasks according to the "reuse" probability). Since it can reuse any previous task's corresponding layer, the number of reuse increases as the task increases. "**adaptation**" expands the network by adding a small number of parameters. Specifically, we add some new additional neural units to the original layer to increase the capacity of the current continual learner. "**new**" spawns new parameters of exactly the same size as that of the current layer's parameters, and all the new parameters are randomly initialized. "**fuse**" makes the new task use the average parameter values of all previous tasks' corresponding layers. For each layer, the agent also needs to decide whether to use "**mask**" to protect some units so that its previous parameters (knowledge) will not be updated and forgotten (see Eq. (2) for details). Figure 1(A) shows the schematic diagram of a continual learner composed of these actions.

We use the convolution neural network as an example to introduce how each action works. For the $l$-th layer of the convolutional neural network, the default kernel size is $5 \times 5$. The action "reuse" (or "fuse") is to use a previous task's $l_{th}$ layer's existing weights (or the average of all previous tasks' $l$-th layers' existing weights) for the $l_{th}$ layer of the current continual learner. For "adaptation", we use a $1 \times 1$ convolution layer added to the original $5 \times 5$ convolution layer in parallel. For the "new" action, we introduce a replicated $5 \times 5$ layer that is initialized randomly. Given the output $y_l^t$ of the

---

[2]In this paper, we use ResNet18 pre-trained on ImageNet in Pytorch to calculate the task similarity.

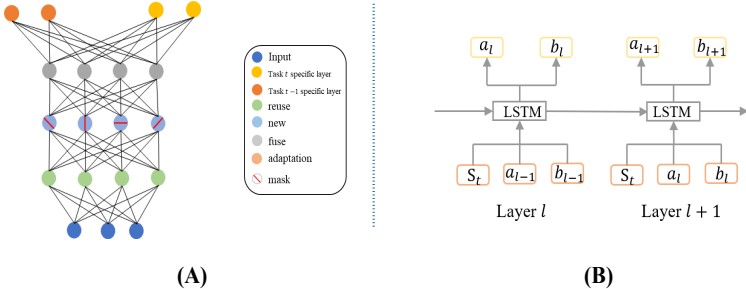

**(A)**             **(B)**

Figure 1: **(A)**: The schematic diagram of the continual learner after the agent learned task $t$. **(B)**: Agent of BNS, where $S_t$ is the environment state of the learning task $t$ ; $a_l$ is the layer-wise action of layer $l$ of the continual learner; $b_l$ is the element-wise action of layer $l$ of the continual learner.

units of the $l$-th layer, our element-wise "mask" $m_l^t$ and the layer final output $y_l^t$ are

$$m_l^t = \sigma(\gamma e_l^t),$$
$$y_l^t = m_l^t \odot y_l^t. \tag{2}$$

where $\sigma$, $\gamma$ and $e_l^t$ are the sigmoid function, a hyper-parameter and the $l_{th}$ layer embedding of task $t$, respectively. It can also be seen from Eq. (2) that "mask" can generate task specific features. From another perspective, the five actions defined for each layer of the continual learner can be divided into two categories: changing the network structure (*i.e.,* "adaptation", "mask") and initializing the parameters (*i.e.,* "reuse", "new" and "fuse").

**Reward:** Once a continual learner is built in an iteration, it is trained with the current task training data $D_{train}^t$ and then a reward is computed for the model of the continual learner for the iteration. We design the reward by considering both the current task $t$'s performance improvement over the previous iteration computed using a Up-ratio function on its validation data and the forgetting rate of all previous tasks using a Drop-ratio function. The Up-ratio of the continual learner constructed by the agent in the current iteration $i$ of action sampling is as follows,

$$u_i^t = \frac{acc_{i,t}^t}{acc_{i-1,t}^t} \tag{3}$$

where $i$, $acc_{i,t}^t$ ($acc_{i-1,t}^t$) are the sample step or iteration of the agent and the accuracy of the current (previous) iteration continual learner $f(\cdot, \theta_t)$ on the current task $t$'s validation set $D_{valid}^t$. The Drop-ratio, which measures $f(\cdot, \theta_t)$'s remembering (CF avoidance) ability, is as follows

$$d_i^t = \frac{1}{t-1} \sum_{k=1}^{t-1} \frac{acc_{i,k}^t}{acc_k^*} \tag{4}$$

where $acc_k^*$ (or $acc_{i,k}^t$) is the accuracy of the final continual learner $f(\cdot, \theta_k)$ (or the current $f(\cdot, \theta_t)$) on the previous task $k$'s validation set $D_{valid}^k \in \mathcal{T}$. Our reward for the current iteration $i$ is

$$r_i^t = d_i^t + \beta u_i^t \tag{5}$$

where $\beta$ is a hyper-parameter. For simplicity, we omit the subscript $i$ from now on.

## 3.2 Training BNS

Figure 1(B) gives our LSTM based agent network. The LSTM samples actions via softmax classifiers in an auto-regressive fashion: the action from the previous step and environment state $S_t$ are fed into the next step. In the first step, the agent network receives an empty action embedding with state $S_t$ as input. BNS has two sets of learnable parameters: the parameters of the agent LSTM $\pi_w(S_t, a_t)$, denoted by $w$, and the parameters of the current continual learner $f(\cdot, \theta_t)$, denoted by $\theta_t$. The training procedure of BNS consists of two interleaving phases. The first phase trains $\theta_t$, the parameters of the current learner, using a whole pass over the training dataset $D_{train}^t$. The second phase trains $w$, the parameters of the agent LSTM. These two phases are alternated during the training of BNS and trained for a fixed number of steps, 200 for each task in our experiments. More details are as follows.

Table 1: Datasets details. For each of the first three datasets, the number of classes is the total number of classes in the dataset. For each of the last two datasets, the number of classes represents the number of classes per task and each dataset has 10 tasks, not shown in the table.

| Dataset | Classes | Training | Testing |
|---------|---------|----------|---------|
| MNIST | 10 | 60,000 | 10,000 |
| CIFAR10 | 10 | 50,000 | 10,000 |
| CIFAR100 | 100 | 50,000 | 10,000 |
| F-EMNIST | 62 (per task) | 7,673 | 858 |
| F-Celeba | 2 (per task) | 960 | 110 |

**Training the continual learner** $f(\cdot, \theta_t)$**:** In this step, the agent first samples an action sequence which is used to reconstruct a continual learner $f(\cdot, \theta_t)$ for learning the current task. Then, we fix the agent's policy $\pi_w(S_t, a_t)$ and minimize the expected loss

$$\mathcal{L}(\theta_t) = \frac{1}{n_t} \sum_{j=1}^{n_t} \ell_t(f(x_j^t, \theta_t), y_j^t) \tag{6}$$

where, $\ell_t$ is the standard cross-entropy loss, computed on a minibatch of the training data, with a model $f(\cdot, \theta_t)$ sampled from $\pi_w(S_t, a_t)$. Our policy is to output an action for each layer of the current continual learner. When the action is "reuse" or "fuse", in order to ensure that these parameters will not change too much and thus alleviate forgetting when training the current task, we introduce L2 transfer regularization [12, 24] to constrain the training of the corresponding layer's parameters. In particular, we mark $\mu^*$ as the parameters obtained from the knowledge repository $\mathcal{K}$ when using the action "reuse" or "fuse". Finally, we perform stochastic gradient descent (SGD) on $\theta_t$ to optimize the following loss function to train each task

$$\mathcal{L}(\theta_t) = \frac{1}{n_t} \sum_{j=1}^{n_t} \ell_t(f(x_j^t, \theta_t), y_j^t) + \eta \cdot ||\theta_t - \mu^*||_2^2 \tag{7}$$

where $\eta$ is a hyper-parameter.

**Training the agent:** When the continual learner $f(\cdot, \theta_t)$ converges, we use Eq. (5) to calculate the reward $r^t$ ($r_i^t$ with iteration $i$ omitted), and then use the standard policy gradient algorithm to update the agent's parameters. In this step, we fix $\theta_t$ and only update the policy parameters $w$, aiming to maximize the expected reward. Formally, we update the parameters $w$ using the following function,

$$w \leftarrow w + \alpha \cdot \bigtriangledown_w log\pi_w(S_t, a_t) \cdot r^t \tag{8}$$

where $\alpha$ is the learning rate of the agent.

**Deriving the continual learner for the current task:** We first sample several models from the trained policy $\pi_w(S_t, a_t)$. For each sampled model, we first train it on the current task and then compute its reward using the current task's validation set and replay buffer $\mathcal{T}$. We then take only the model with the highest reward as the final learned model.

## 4   Experiments

We evaluate the proposed BNS[3] on five image classification datasets, three standard benchmarks MNIST [28], CIFAR10 and CIFAR-100 [26] and two additional datasets F-EMNIST and F-Celeba [21] originally used for federated learning, which have similar tasks and thus allow us to evaluate the knowledge transfer ability of BNS. F-EMNIST has 10 tasks and each task contains the written digits and characters (62 classes) of one individual writer. F-Celeba also has 10 tasks and each of them contains images of a celebrity labeled by whether he/she is smiling or not. The datasets statistics are given in Table 1. Our goal is two-fold: **(1)** to verify whether searching a specific network for each task through reinforcement learning can deal with CF, and **(2)** to verify whether similarities between tasks can help build a good network for knowledge transfer to achieve improved accuracy.

---

[3]The code of BNS can be found at: `https://github.com/lalalaup6/BNS`

## 4.1 Baselines

We consider the following 9 baselines for comparison with BNS, which is a task continual learning (Task-CL) system. **SGD** uses a feature extractor and a classifier as the continual learner to incrementally learn all tasks. There is no mechanism for dealing with CF or knowledge transfer. **LWF** [33] uses knowledge distillation to deal with CF. **EWC** [63] is a commonly used baseline in most continual learning papers. **IMM** [30] combines the sequentially trained independent models for different tasks to perform all the tasks in the sequence. **HAT** [50] uses masks to protect task parameters to deal with CF. **GEM** [36] is a replay method for Task-CL. **UCL** [2] is a Task-CL method that tries to improve HAT. **HNET+R** [56] is a latest replay Task-CL method (HNET+R performs better than HNET). **GPM** [48] is a recent Task-CL method that learns new tasks by taking gradient steps in orthogonal direction to the gradient subspaces deemed important for the past tasks. Note, LWF, EWC and IMM were originally designed for class continual learning (Class-CL). They were adapted for Task-CL in HAT [50]. We use these versions and run their code in the HAT repository.

## 4.2 Implementation Details and Evaluation Metrics

For experiments using MNIST and F-EMNIST, our BNS and all baselines except HNET+R[4] use the same three fully connected layers as the shared feature extractor, and a classifier on top with one fully connected layer for each task. In these two datasets, BNS does not use the action "adaptation" in agent to sample actions for the continual learner. For experiments with F-Celeba, CIFAR10 and CIFAR100, BNS and all baselines except HNET+R use AlexNet [27] as the basic network, which is borrowed from HAT [50]. Its shared feature extractor consists of three convolutional layers followed by two fully connected layers. The task specific classifier is one fully connected layer. In these three datasets, BNS uses all actions to search for a suitable architecture for the continual learner. For all experiments, we use a single Nvidia RTX 2080Ti GPU.

**Parameter settings:** For BNS, we use SGD as the optimizer with learning rate $0.1$ to train the continual learner $f(\cdot, \theta_t)$ except F-EMNIST and F-Celeba (learning rate $0.01$). The parameters of the agent (LSTM) is updated by the Adam optimizer using the learning rate $0.0001$. The hyperparameters $\eta$ and $\beta$ are set to $0.001$ and $0.003$ respectively. To be consistent with the baseline settings, our continual learner trains 100 epochs for all datasets except MNIST (10 epochs). We use ten percent of the training data of each task as the validation set for reward computation. After a task is learned, its validation set is saved in the replay buffer.

**Evaluation metrics**: We adopt the final classification accuracy (ACC) as the basic evaluation metric. For each dataset, we report the average accuracy of all tasks tested using their respective test data after all tasks have been learned, thus the final accuracy. In order to compare the ability in overcoming CF and knowledge transfer, we introduce two other indicators. The first one is backward transfer (**BWT**) [36] to measure forgetting, which indicates how much the learning of new tasks has influenced the accuracy of the models for the previous tasks. BWT $< 0$ means that the learning of new tasks leads to some forgetting, BWT $> 0$ indicates that the learning of new tasks helps the previous tasks. The second one is a new metric (**Trans**), which measures the forward transfer ability, showing how helpful the knowledge learned from previous tasks is to each new task in learning. In particular, Trans compares the accuracy of the final continual learner on each task with the accuracy of training an independent model for each task separately (no continual learning).

$$\text{ACC} = \frac{1}{N}\sum_{t=1}^{N}\text{acc}_{N,t}; \quad \text{BWT} = \frac{1}{N-1}\sum_{t=1}^{N-1}\text{acc}_{N,t} - \text{acc}_{t,t}; \quad \text{Trans} = \frac{1}{N}\sum_{t=1}^{N}\text{acc}_{N,t} - \text{acc}_t^{'}. \quad (9)$$

where $N$ is the total number of tasks learned; $\text{acc}_{t,t}$ is the accuracy on the test set of task $t$ right after task $t$ is learned; $\text{acc}_t^{'}$ is the accuracy on the test set of task $t$ of the independent model trained on task $t$. Accordingly, $\text{acc}_{N,t}$ is the accuracy of task $t$ after all $N$ tasks have been learned. Note that since our BNS is built on HAT's architecture and may also sample the action "mask", the key mechanism in HAT for protecting the sub-network of each task to prevent forgetting. So comparing with HAT, we will see the knowledge transfer effect of BNS compared to a Task-CL system.

---

[4]Experiments with HNER+R use its own network as it is highly complex and is very difficult to change.

Table 2: Average results of ACC, BWT and Trans for BNS and baselines on MNIST, CIFAR10 and CIFAR100 with dissimilar tasks over 5 runs, where $A_5$, $A_2$ and $A_{50}$ represent the number of tasks that are continually learned on the corresponding dataset. Standard deviation, number of model parameters, and execution time are given in Appendix A.3.

| Model | MNIST $A_5$ | | | MNIST $A_2$ | | | CIFAR10 $A_5$ | | | CIFAR10 $A_2$ | | | CIFAR100 $A_{50}$ | | |
|---|---|---|---|---|---|---|---|---|---|---|---|---|---|---|---|
| | ACC | BWT | Trans | ACC | BWT | Trans | ACC | BWT | Trans | ACC | BWT | Trans | ACC | BWT | Trans |
| SGD | 85.77 | −17.45 | −14.06 | 90.47 | −17.26 | −8.83 | 73.57 | −23.46 | −18.83 | 76.03 | −23.28 | −10.58 | 61.14 | − −30.24 | −29.22 |
| LWF | 99.53 | −0.36 | −0.28 | 98.82 | −0.46 | −0.48 | 76.86 | −18.63 | −14.95 | 83.22 | −6.62 | −3.40 | 52.91 | −37.90 | −37.44 |
| EWC | 85.99 | −17.23 | −13.82 | 91.58 | −15.34 | −7.71 | 79.44 | −13.63 | −12.37 | 84.18 | −3.58 | −2.43 | 68.78 | −21.88 | −21.57 |
| IMM | 87.25 | −15.61 | −12.56 | 94.46 | −9.17 | −4.83 | 72.81 | −22.09 | −19.00 | 78.39 | −16.58 | −8.23 | 67.86 | −22.53 | −22.48 |
| UCL | 98.74 | −1.22 | −1.07 | 97.40 | −0.67 | −1.90 | 85.29 | −5.93 | −6.52 | 83.66 | −3.04 | −2.95 | 64.28 | −25.54 | −26.21 |
| GPM | 98.56 | −0.65 | −1.25 | 98.27 | −0.57 | −1.03 | 88.62 | −2.68 | −3.19 | 83.70 | −4.72 | −2.92 | 78.43 | −3.76 | −11.92 |
| GEM | 99.28 | −0.36 | −0.54 | 96.45 | −0.78 | −2.84 | 77.64 | −4.03 | −14.17 | 62.87 | −2.09 | −23.74 | 78.05 | −0.78 | −12.30 |
| HNET+R | 99.61 | −0.21 | −0.20 | 98.58 | −0.58 | −0.72 | **92.17** | −1.26 | **0.31** | 85.76 | −0.18 | −0.42 | 76.73 | −15.17 | −13.72 |
| HAT | 99.74 | 0.00 | −0.08 | 99.03 | 0.00 | −0.26 | 90.89 | 0.00 | −0.92 | 86.25 | 0.00 | −0.36 | 80.46 | **0.00** | −9.89 |
| BNS | **99.87** | **0.01** | **0.04** | **99.26** | **0.00** | **−0.04** | 91.40 | **0.67** | **−0.42** | **87.64** | **0.01** | **1.02** | **82.39** | −4.35 | **−7.96** |

Table 3: Average results of ACC, BWT and Trans for BNS and the baselines on F-EMNIST and F-Celeba with similar tasks over 5 runs.

| Model | F-EMNIST | | | F-Celeba | | |
|---|---|---|---|---|---|---|
| | ACC | BWT | Trans | ACC | BWT | Trans |
| SGD | 13.49 | −26.66 | −25.87 | 54.93 | −1.62 | −6.88 |
| LWF | 8.11 | −2.20 | −31.25 | 57.18 | −1.61 | −4.63 |
| EWC | 43.47 | −8.67 | 4.10 | 55.27 | −1.22 | −6.53 |
| IMM | 17.52 | −25.74 | −21.84 | 57.81 | −0.80 | −4.00 |
| UCL | 49.75 | −1.63 | 10.38 | 55.26 | −5.65 | −6.55 |
| GPM | 57.84 | 6.40 | 19.07 | 58.08 | **1.03** | −3.69 |
| GEM | 53.82 | **21.75** | 15.02 | 54.51 | −3.62 | −7.27 |
| HNET+R | 24.95 | −43.69 | −13.85 | 55.84 | −3.23 | −5.92 |
| HAT | 56.02 | 0.00 | 16.65 | 59.82 | 0.00 | −1.99 |
| BNS | **60.38** | 0.53 | **21.01** | **65.44** | −2.59 | **3.62** |

## 4.3 Experimental Results

To evaluate the effectiveness of forgetting avoidance and knowledge transfer of the continual learner of BNS, we use the 3 metrics in Eq. 9. Each row in the table gives the average results over 5 runs of all compared systems. The mean of results are summarized in Table 2 and Table 3. The standard deviations are summarized in Tables 6 and 7 of Appendix A.3. For MNIST, CIFAR10 and CIFAR100, we evaluate the effectiveness under three settings, 5 tasks ($A_5$), 2 tasks ($A_2$) and 50 tasks ($A_{50}$). For example, CIFAR10 $A_5$ means that the original dataset is divided into 5 tasks and each task contains two classes. For F-MNIST and F-Celeba, the 10 tasks are fixed in the original datasets. We analyze the results in the two tables from the following perspectives.

**Final accuracy (ACC):** BNS performs very well. Only HNET+R performs slightly better than BNS on CIFAR10 $A_5$. However, HNET+R performs poorly in the other experiments in Table 2, especially on CIFAR100 $A_{50}$. On the two similar task datasets F-EMNIST and F-Celeba (Table 3), the current state-of-the-art models, HNET+R and GPM, are very weak compared to BNS. They are even poorer than HAT. BNS outperforms all baselines by a large margin, which shows that its ability to handle multiple tasks using the final continual learner is powerful.

**With and without continual learning (CL):** Here we compare the results of learning each task separately (without CL) and CL and study the impact of CL. The *Trans* metric is for this purpose. Recall Trans is the difference between the final accuracy ACC of the continual learner after all tasks are learned and the accuracy of learning a model for each task separately and independently (which is ACC - Trans). The Trans column in Table 2 shows that BNS produces similar or better results than learning separately when the tasks are different. The baselines are clearly poorer. For CIFAR100 $A_{50}$, there are some loss of accuracy, but the baselines lose even more. The Trans column in Table 3 shows a very different story when we have similar tasks. We see that five CL baselines can improve the accuracy (positive Trans values) of learning separately without CL (ACC - Trans) for F-EMNIST, which show some knowledge transfer. But the knowledge transfer ability of BNS is significantly better than all baselines. For example, the Trans values of GPM on F-EMNIST is $19.07\%$ and on F-Celeba is $-3.69\%$ (negative transfer), but these of BNS are $21.01\%$ and $3.62\%$ respectively. Note that the final accuracy ACC of all other baselines are weaker than HAT, which is markedly poorer than BNS. This again demonstrates the knowledge transfer ability of BNS.

**Overcoming catastrophic forgetting (CF):** BWT measures the ability of preventing CF. SGD performs extremely poorly with very negative values (negative backward transfer or CF) for BWT for

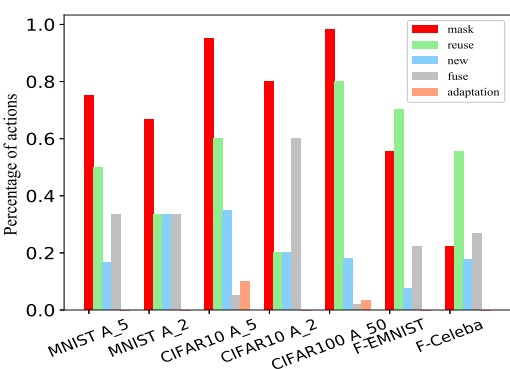

Figure 2: The percentage of each action used in the continual learner. Note that "mask" co-exists with other actions. At the same time, "adaptation" is accompanied by "reuse".

all datasets. This is because SGD has no mechanism to deal with CF. Comparing the BWT values of BNS and the baselines, we can see that BNS performs very well except a few cases with GPM and HAT (no CF), but their final accuracy ACC values are substantially lower.

**Backward knowledge transfer:** As indicated in Section 4.2, BWT not only measures CF avoidance (*i.e.*, BWT < 0), but also measures backward knowledge transfer from new tasks to old tasks (*i.e.*, BWT > 0). For all datasets, HAT has no backward transfer (BWT = 0.00) and no forgetting. Our BNS achieves none negative backward transfer in 5 out of the 7 experiments. The baseline GPM achieves only in 3, out of which 2 (F-EMNIST and F-CEleba) still result in much lower final accuracy (ACC) than BNS, indicating that the forward transfer capability of GPM is poor (see below).

**Forward knowledge transfer:** This is defined as the difference between the accuracy of a task when it is first learned (ACC $- \frac{N-1}{N}$BWT) and the accuracy of learning the task separately (ACC $-$ Trans), *i.e.*, Trans $- \frac{N-1}{N}$BWT, which shows whether the knowledge learned from previous tasks can help learn the new task. We summarize the experimental results of this metric in Appendix A.2. We can again see that BNS is substantially better overall. As discussed above, for the two similar tasks datasets in Table 3, GPM's backward transfer is more effective, but GPM's *forward transfer* (Trans $- \frac{N-1}{N}$BWT) results are only $13.31\%$ and $-4.61\%$ but those of BNS are $20.53\%$ and $5.95\%$ respectively, which explains why GPM's final accuracy ACC results are much lower than BNS. For the best baseline HAT, they are respectively $16.65\%$ and $-1.99\%$, which are also substantially lower than those of BNS.

In summary, we can conclude that for dissimilar tasks with little shared knowledge, BNS can deal with CF as well as state-of-the-art baselines and in many cases, it can even achieve some positive knowledge transfer. For similar tasks with shared knowledge, the knowledge transfer capability of BNS is considerably better than the baselines. This indicates that BNS is a much better model for achieving the two objectives of Task-CL, CF avoidance and knowledge transfer.

## 4.4 Actions Used by the Agent

In Figure 2, we plot the percentage of each action used in the action sequences generated by the agent for all tasks in our experiments. It can be seen that in the datasets with similar tasks (F-EMNIST and F-Celeba), the percentage of action "masks" used is much lower. This is so because for similar tasks sharing and knowledge transfer are needed but masks are for protecting previous knowledge. In Appendix A.1, we will present a case study to show the action selection of the agent.

## 4.5 Ablation Experiments and Analysis

As discussed earlier, BNS produces a task-related network structure and gives a specific initialization method for each task according to the similarity of the task to previous tasks. In order to verify the effectiveness of including the task similarity, we compare the results of three settings on five datasets MNIST $A_5$, CIFAR10 $A_5$, CIFAR100 $A_{50}$, F-EMNIST and F-Celeba. The first setting is without task similarity as the agent input, which degenerates into a traditional neural architecture search algorithm,

Table 4: Ablation study. BNS (zero-smi) means that no task similarity is used as an input to the agent. BNS (one-smi) (or BNS) means that only the cosine similarity (or cos, KL and $L_p$-norm together) is (are) used to calculate task correlation and is used as an input to the agent.

| Components | MNIST | CIFAR10 | CIFAR100 | F-EMNIST | F-Celeba |
|---|---|---|---|---|---|
| SGD | 85.77 | 73.57 | 61.14 | 13.49 | 54.93 |
| BNS (zero-sim) | 99.24 | 88.99 | 78.50 | 57.75 | 62.28 |
| BNS (one-sim) | 99.85 | 90.73 | 80.21 | 58.41 | 66.62 |
| BNS | 99.87 | 91.40 | 82.39 | 60.38 | 65.44 |

referred to as BNS (zero-sim). The second setting uses one similarity, cosine similarity, as the agent input, referred to as BNS (one-sim). The third setting uses all similarity measures as an input to the agent, which is our final BNS model. Table 4 gives the results of the ablation experiments.

By comparing the results of BNS (zero-sim) and SGD, it can be seen that after storing the parameters learned in the past, using NAS (neural architecture search) to search for each task has a significant positive effect on overcoming catastrophic forgetting (CF). That is, designing a new initialization method for each task using the parameters of previous tasks and appropriately changing the network capacity can already help overcome CF significantly. Additionally, introducing one or more similarities as the input to the agent further improves the model's ability to overcome CF (see the results of BNS (one-sim) and BNS). This demonstrates that it is highly effective to generate the network structure and initialization method based on task similarity.

## 5  Conclusion

This paper proposed a novel reinforcement learning based method, BNS, for task continual learning (Task-CL). The proposed model can dynamically adjust the network structure and parameter initialization method of each task by exploiting the similarities and differences of the tasks to achieve both objectives of Task-CL, overcoming catastrophic forgetting and transferring knowledge across tasks to improve accuracy. Experimental results showed that the performance of BNS is superior to the state-of-the-art baselines in both forgetting prevention and knowledge transfer. The knowledge transfer ability of the baselines are especially weak compared to BNS. One limitation of work is that it requires a large amount of memory and compute. In our future work, we plan to try a generative approach to generate the parameters of the past models.

## Acknowledgments

This work was supported in part by an internal grant of Peking University and National Key R&D Program of China 2020AAA0106600.

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
