# A Appendix

## A.1 Case study of a learned continual learner

By analyzing the continual learner built by the agent for each task, we can make some additional observations. As an illustration, we show the BNS continual learner built by the agent for each task of MNIST $A_5$ in Figure 3 . We notice that when learning a new task, the first layer (*i.e.,* the lowest layer) of the continual learner tends to "reuse" the corresponding layer's parameters of a previous task and use the element-wise action "mask" (indicated by a red line in each unit or neuron). Generally, the lower layers tend to extract some basic features, and these features can be continually shared among multiple tasks. At the same time, because each task has its specific and different characteristics, the lower layers of the structure also tend to use the "mask" action to control the output value of each dimension to generate task-specific features suitable for the current task. The third layer of the continual learner tends to be the parameters of all previous tasks and do not use element-wise action "mask", which shows that the higher levels have the ability to combine low-dimensional features. Therefore, there are more operations to "fuse" to combine the abilities of previous tasks.

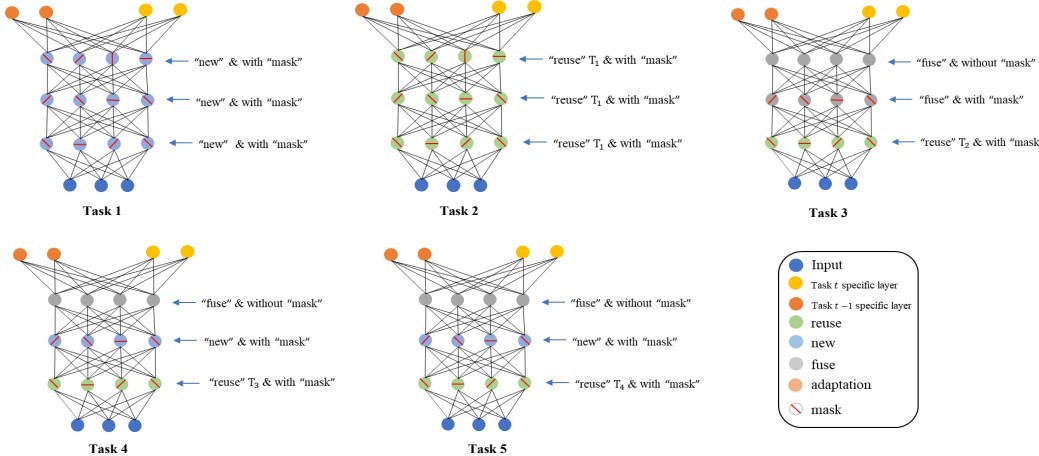

Figure 3: The continual learner of each task for MNIST $A_5$, where "reuse" $T_i$ indicates that the corresponding layer's parameters of task $i$ is used as the initialization parameters for the new task. With/without "mask" means use (or don't use) the "mask" action.

## A.2 Forward Transfer Analysis

Here, we introduce another forward transfer metric (**F-ACC**), which indicates how helpful the knowledge learned from previous tasks is to the new task.

$$\text{F-ACC} = \frac{1}{N}\sum_{t=1}^{N}\text{acc}_{t,t} - \text{acc}_t^{'} = \text{Trans} - \frac{N-1}{N}BWT \tag{10}$$

where $\text{acc}_{t,t}$ is the classification accuracy on the test dataset of task $t$ right after task $t$ is learned, $\text{acc}_t^{'}$ is the accuracy on the test set of task $t$ of the independent model trained on task $t$ using a separate network of the same size. The results are summarized in Table 5. It can be seen from Table 5 that our BNS is better overall than the baseline models in terms of forward transfer. Although in some cases, some baselines' F-ACC scores are better but their final accuracy results are poorer due to forgetting (see Table 2 in the paper).

## A.3 Standard deviation, number of model parameters, and execution time

This section reports the standard deviations of the experimental results given in the paper, the number of parameters of each model and the running time on each dataset. Table 6 and 7 report the standard deviations of all experiments running 5 times for the results in Table 2 and Table 3 in the paper. Table

Table 5: Results of F-ACC for our BNS and all baseline models on all datasets. They are the averages of 5 random runs.

| Model | MNIST $A_5$ | MNIST $A_2$ | CIFAR10 $A_5$ | CIFAR10 $A_2$ | CIFAR100 $A_{50}$ | F-EMNIST | F-Celeba |
|---|---|---|---|---|---|---|---|
| SGD | $-0.08$ | $-0.19$ | $-0.06$ | $1.06$ | $0.40$ | $-1.87$ | $-5.42$ |
| LWF | $0.01$ | $-0.25$ | $-0.04$ | $-0.08$ | $-0.30$ | $-29.27$ | $-3.18$ |
| EWC | $-0.04$ | $-0.05$ | $-1.46$ | $-0.64$ | $-0.13$ | $11.90$ | $-5.43$ |
| IMM | $-0.08$ | $-0.24$ | $-1.34$ | $0.06$ | $-0.40$ | $1.33$ | $-3.28$ |
| UCL | $-0.09$ | $-1.56$ | $-1.77$ | $-1.43$ | $-1.18$ | $11.84$ | $-1.47$ |
| GPM | $-0.73$ | $-0.74$ | $-1.04$ | $-0.55$ | $-8.23$ | $13.31$ | $-4.61$ |
| GEM | $-0.25$ | $-2.44$ | $-10.94$ | $-22.69$ | $-11.53$ | $-4.55$ | $-4.01$ |
| HNET+R | $-0.03$ | $-0.43$ | $1.32$ | $-0.33$ | $1.15$ | $25.48$ | $-3.01$ |
| HAT | $-0.07$ | $-0.26$ | $-0.92$ | $-0.36$ | $-9.89$ | $16.65$ | $-1.99$ |
| BNS | $0.04$ | $-0.04$ | $-0.95$ | $1.02$ | $-3.69$ | $20.53$ | $5.95$ |

8 lists the number of network parameters of each model in each experiment. Because the number of model parameters of each task of our BNS method changes during the learning process, Therefore, for each baseline, we calculate the model parameters of all tasks in each experimental setting to obtain the results in Table 8. *Other Baselines* in Table 8 include SGD, LWF, EWC, IMM, UCL, GPM and GEM. In the main paper, we mentioned that all models except HNET+R use the same backbone, so most of the baseline parameters are exactly the same. Since HAT will train some task embeddings for each task separately, the parameter number of HAT is slightly different from other baselines. We report the number of parameters for our BNS system with (BNS) and without (BNS (w/o LSTM)) the parameters of the LSTM in the agent. Table 9 is the elapsed training time of each model in each experiment. It can be observed from Table 9 that our RL algorithm requires a relatively longer time to train. This is due to the use of reinforcement learning.

Table 6: Standard deviations of ACC, BWT and Trans for BNS and baselines on MNIST, CIFAR10 and CIFAR100 with dissimilar tasks over 5 runs, where $A_5$, $A_2$ and $A_{50}$ represent the number of tasks that are continually learned using the corresponding dataset.

| Model | MNIST $A_5$ | | | MNIST $A_2$ | | | CIFAR10 $A_5$ | | | CIFAR10 $A_2$ | | | CIFAR100 $A_{50}$ | | |
|---|---|---|---|---|---|---|---|---|---|---|---|---|---|---|---|
| | ACC | BWT | Trans | ACC | BWT | Trans | ACC | BWT | Trans | ACC | BWT | Trans | ACC | BWT | Trans |
| SGD | 0.57 | 0.73 | 0.57 | 1.83 | 3.87 | 1.83 | 1.97 | 1.26 | 1.97 | 0.66 | 1.38 | 0.66 | 2.88 | 2.96 | 2.88 |
| LWF | 0.04 | 0.05 | 0.04 | 0.41 | 0.23 | 0.41 | 2.08 | 2.71 | 2.08 | 1.89 | 3.46 | 1.89 | 0.49 | 0.80 | 0.49 |
| EWC | 0.53 | 0.68 | 0.53 | 1.07 | 2.13 | 1.07 | 1.06 | 1.67 | 1.06 | 0.51 | 1.81 | 0.51 | 3.05 | 3.14 | 3.05 |
| IMM | 0.20 | 0.26 | 0.20 | 1.96 | 3.90 | 1.96 | 0.94 | 0.68 | 0.94 | 0.75 | 1.41 | 0.75 | 0.11 | 0.12 | 0.11 |
| UCL | 0.34 | 0.38 | 0.34 | 0.63 | 0.35 | 1.90 | 1.93 | 2.87 | 1.93 | 0.64 | 0.32 | 0.64 | 0.43 | 0.59 | 0.43 |
| GPM | 0.11 | 0.07 | 0.11 | 0.16 | 0.08 | 0.16 | 0.48 | 0.36 | 0.48 | 0.48 | 0.27 | 0.48 | 0.84 | 0.32 | 0.84 |
| GEM | 0.12 | 0.08 | 0.12 | 0.58 | 0.22 | 0.58 | 1.26 | 0.96 | 1.26 | 1.69 | 0.71 | 1.69 | 1.17 | 0.10 | 1.17 |
| HNET+R | 0.10 | 0.11 | 0.10 | 0.23 | 0.39 | 0.23 | 1.28 | 1.63 | 1.28 | 0.86 | 1.07 | 0.86 | 1.30 | 1.57 | 1.30 |
| HAT | 0.08 | 0.00 | 0.08 | 0.06 | 0.00 | 0.06 | 0.99 | 0.00 | 0.99 | 0.51 | 0.00 | 0.51 | 0.55 | 0.00 | 0.55 |
| BNS | 0.04 | 0.02 | 0.04 | 0.03 | 0.00 | 0.03 | 0.48 | 0.47 | 0.48 | 0.13 | 0.01 | 0.13 | 0.68 | 0.57 | 0.68 |

Table 7: Standard deviations of ACC, BWT and Trans for BNS and the baselines on F-EMNIST and F-Celeba with similar tasks over 5 runs.

| Model | F-EMNIST | | | F-Celeba | | |
|---|---|---|---|---|---|---|
| | ACC | BWT | Trans | ACC | BWT | Trans |
| SGD | 7.78 | 9.29 | 7.78 | 3.59 | 2.67 | 3.59 |
| LWF | 1.22 | 2.71 | 1.22 | 1.36 | 2.27 | 1.36 |
| EWC | 7.14 | 4.02 | 7.14 | 2.59 | 2.22 | 2.59 |
| IMM | 3.07 | 3.29 | 3.07 | 2.34 | 2.81 | 2.34 |
| UCL | 2.96 | 4.48 | 2.96 | 1.06 | 2.60 | 1.06 |
| GPM | 2.44 | 0.69 | 2.44 | 2.69 | 0.93 | 2.69 |
| GEM | 1.60 | 1.81 | 1.60 | 0.02 | 1.45 | 0.02 |
| HNET+R | 0.94 | 2.39 | 0.94 | 0.88 | 2.89 | 0.88 |
| HAT | 0.22 | 0.00 | 0.22 | 2.65 | 0.00 | 2.65 |
| BNS | 0.29 | 0.35 | 0.29 | 1.16 | 2.56 | 1.16 |

Table 8: Number of parameters of all baselines and our BNS system. Each number is the sum of all task model parameters in each setting in the final network after all tasks have been learned.

| Model | MNIST $A_5$ | MNIST $A_2$ | CIFAR10 $A_5$ | CIFAR10 $A_2$ | CIFAR100 $A_{50}$ | F-EMNIST | F-Celeba |
|---|---|---|---|---|---|---|---|
| Other Baselines | 957.6k | 383.0k | 4619.7k | 1847.8k | 47119.2k | 2406.2k | 9259.9k |
| HAT | 960.6k | 384.7k | 4621.9k | 1848.8k | 47141.9k | 2412.2k | 9264.5k |
| HNET+R | 759.3k | 304.1k | 1818.2k | 738.8k | 20962.6k | 1380.3k | 3363.2k |
| BNS (w/o LSTM) | 959.4k | 383.4k | 4625.2k | 1847.9k | 47135.3k | 2245.7k | 9261.1k |
| BNS | 2077.7k | 1501.6k | 5743.5k | 2966.2k | 48253.3k | 3363.9k | 10379.4k |

Table 9: Training time (minutes) used by our BNS model and all baselines in each experiment.

| Model | MNIST $A_5$ | MNIST $A_2$ | CIFAR10 $A_5$ | CIFAR10 $A_2$ | CIFAR100 $A_{50}$ | F-EMNIST | F-Celeba |
|---|---|---|---|---|---|---|---|
| SGD | 1.96 | 1.75 | 8.67 | 9.84 | 21.66 | 1.33 | 0.36 |
| LWF | 2.63 | 1.65 | 13.32 | 12.06 | 38.85 | 1.58 | 0.45 |
| EWC | 3.65 | 2.04 | 18.84 | 16.75 | 105.2 | 3.81 | 0.64 |
| IMM | 6.41 | 4.13 | 13.46 | 14.59 | 78.49 | 3.28 | 0.54 |
| HAT | 2.34 | 1.87 | 16.59 | 20.22 | 28.69 | 2.63 | 1.06 |
| GPM | 6.67 | 6.51 | 28.42 | 38.14 | 130.47 | 1.91 | 2.38 |
| GEM | 5.07 | 3.22 | 706.20 | 459.68 | 219.94 | 35.03 | 23.69 |
| UCL | 1.41 | 2.78 | 45.46 | 42.42 | 56.73 | 3.45 | 1.93 |
| HNET+R | 6.51 | 1.82 | 52.09 | 43.60 | 192.58 | 23.35 | 1.72 |
| BNS | 410.65 | 189.61 | 5542.62 | 3172.67 | 12988.14 | 87.38 | 65.21 |