# OpenReview forum: "BNS: Building Network Structures Dynamically for Continual Learning"
_NeurIPS.cc/2021/Conference — NeurIPS 2021 Poster_

### Official Review · Reviewer_t7Yn · 2021-07-16

**Rating:** 6
**Confidence:** 3

**Summary:**

This paper studies continual learning when the model is required to fit a sequence of tasks (with task identifiers) and maximize the average performance on all tasks. The goal is to prevent both catastrophic forgetting (CF), and improve forward knowledge transfer. It presents a neural architecture search (NAS) approach, namely BNS, to train an agent to design the new architecture for each new task. The idea of applying NAS in lifelong learning setting is interesting and novel, and the experimental results do show performance gains.

**Limitations And Societal Impact:**

See above.

**Main Review:**

Overall, I think the paper is novel with good results. But there are certain technical details that are not immediately clear to me (and the link to code is empty), that I would appreciate if the author can clarify a bit in the rebuttal, and I am happy to adjust my score based on that.

1. How do you evaluate the score for, say the first task, after training on all tasks? Do you save the best model for it in the memory buffer? Or do you utilize the agent to create a new model? If it's the former, wouldn't that requires linear memory in terms of number of tasks? If it's the later, what is the environment, or the base learner for the input to the agent? Does that mean the agent can utilize knowledge learned in the later task to construct model for earlier tasks?
2. It seems to me that the agent is the most prone to CF. Essentially, it is trained to optimize the current task performance in a lifelong learning setup. Although preventing CF is included as part of the reward, I am not super convinced that this is a signal that is strong enough. Have you tried use other lifelong learning methods to regularize the agent?
3. A big concern of NAS is its efficiency, and this is even more serious the case here since you are learning to search for not just one but many tasks. Given the lifelong learning setting is often used in a online setting in reality, it seems to me that the computational complexity may be prohibitively large for your method. Can you provide the runtime compared with other methods for your experiments?

**Time Spent Reviewing:**

2

---

> ### Author Response · Authors · 2021-08-10
> **Our responses to your comments.**
>
> 1. Re:”How do you evaluate the score for, say the first task, after training on all tasks? Do you save the best model for it in the memory buffer? Or do you utilize the agent to create a new model? If it's the former, wouldn't that requires linear memory in terms of number of tasks? If it's the later, what is the environment, or the base learner for the input to the agent? Does that mean the agent can utilize knowledge learned in the later task to construct model for earlier tasks?”
>
> It is not the former. For each new task, the agent builds a model that is able to cover all tasks learned so far. The original models for the previous tasks are not used in the final evaluation. That is, the model from the last task constructed by the agent is the final model for all tasks (old and new) learned so far and is used to test on (or evaluate the score for) all tasks including the first task.
>
> Specifically, the whole training of our BNS system is done using reinforcement learning (RL). Given a new task, the system starts with computing the environment state, which is a similarity vector storing the similarity of the new task with each previous task. The RL training now starts. In each RL iteration, the agent takes the state (which is fixed for each new task) as the input to generate an action sequence, which is used to create a continual learner network for the new task, including initialization. The continual learner is then trained using the new task data to produce a model. After that, a reward is produced from the trained model of the current continual learner using the Drop-ratio & Up-ratio (which are computed with the validation data of the current task and the saved/replay samples of the previous tasks and the trained model). The reward is then used to train the agent (which is a LSTM). After that, the next RL iteration starts. After the RL converges, the final continual learner is the model for the new task (which performs well on the both the new and all the old tasks).
>
> 2. Re:”It seems to me that the agent is the most prone to CF. Essentially, it is trained to optimize the current task performance in a lifelong learning setup. Although preventing CF is included as part of the reward, I am not super convinced that this is a signal that is strong enough. Have you tried use other lifelong learning methods to regularize the agent?”
>
> As we can see from the BNS training process given in the above response to question 1, the agent (a LSTM) is trained with the reward obtained from the continual learner in the RL learning process. The only training goal of the agent is to generate a good action sequence to be used to create a continual learner network for the new task. Thus, the agent does not need to retain anything from the previous tasks and it thus has no CF issue. In dealing with CF, apart from the reward signal, the “mask”, “adaption”, “reuse”and “fuse”actions are all designed for dealing with CF (and also for knowledge transfer). Regularization (Eq. 7) is used in our approach as well.
>
> 3. Re:”A big concern of NAS is its efficiency, and this is even more serious the case here since you are learning to search for not just one but many tasks. Given the lifelong learning setting is often used in an online setting in reality, it seems to me that the computational complexity may be prohibitively large for your method. Can you provide the runtime compared with other methods for your experiments?”
>
> The running time of each baseline and our BNS system are given Table 9 of Appendix A.3 in Supplementary. Indeed, RL-based network structure search methods usually take a relative long time to train. One of our purposes of using the action "reuse" to reuse the parameters of some previous tasks is to speed up the training convergence. Our current method is not designed for online learning setting.

---

> > ### Comment · Reviewer_t7Yn · 2021-08-22
> > **Thanks**
> >
> > Thank you for your response. Now I understand your method better and I have increased my score.

---

### Official Review · Reviewer_W4Fy · 2021-07-16

**Rating:** 6
**Confidence:** 4

**Summary:**

The paper proposes BNS  for task continual learning, which can dynamically adjust the network structure and parameter initialization method of each task by exploiting the similarities and differences of the tasks to achieve both objectives of Task-CL, overcoming catastrophic forgetting and transferring knowledge across tasks to improve the performance.
The contribution of the paper is to propose a knowledge transfer method across different tasks for continual learning.

**Ethics Review Area:**

["I don’t know"]

**Limitations And Societal Impact:**

1. The main concern is the RL usually needs large trial and error, and thereby low-efficient. The paper should give the learning cost.
2. The number of datasets is relatively small and hard to verify the effectiveness of continual learning.

**Main Review:**

1. The paper address the problem of knowledge transfer across tasks in continual learning, which has received relatively little attention.  This problem is worth to be studied.
2. The paper formulates continual learning as a reinforcement learning problem and defines a novel RL setting.
3. The experiment results show the proposed method is effective.
4. The proposed method combines the structure search and element-wise action to better find the effective subnetwork for each task.

**Time Spent Reviewing:**

3

---

> ### Author Response · Authors · 2021-08-10
> **Our responses to your comments.**
>
> 1. Re:”The main concern is the RL usually needs large trial and error, and thereby low-efficient. The paper should give the learning cost.”
>
> The learning costs are given in Table 9 of Appendix A.3 in Supplementary. Indeed, learning with RL requires a relatively long time. One of the reasons that we use the action "reuse" to reuse the parameters of some previous tasks is to speed up the convergence of the continual learner, thereby alleviating this problem.
>
> 2. Re:”The number of datasets is relatively small and hard to verify the effectiveness of continual learning.”
>
> We used 5 datasets in our experiments (3 datasets with dissimilar tasks (Table 2) and 2 datasets with similar tasks (Table 3)), which is about the same as other CL papers (e.g., references [21, 29, 31, 49, 51, 56, 63] in our paper). The datasets are also standard CL benchmarks. In addition, we also used a large number of tasks, 50 tasks in one setting (Table 2), which is not commonly used in the CL literature.

---

### Official Review · Reviewer_mjUa · 2021-07-16

**Rating:** 6
**Confidence:** 4

**Summary:**

The paper focuses on transfer of knowledge across tasks in a continual learning setting without catastrophic forgetting. The proposed method dynamically builds neural network structures. For each new task an agent samples a sequence of actions that integrates knowledge transfer from the old tasks to construct the continual learner. Old data are stored in a replay buffer, which are used for reward calculation rather than in training. The model thus incrementally learns a sequence of $N$ tasks. The continual learner is trained in a reinforcement learning setting where the reward is a combination of performance increase compared to a previous iteration and the forgetting rate of all previous tasks promoting the accuracy on all previous tasks.


**Ethical Concerns:**

There are no ethical concerns.

**Limitations And Societal Impact:**

Yes, the authors have addressed the limitations of their work in the supplementary material.

**Main Review:**

The paper provides a clear definition of task-continual learning and class-continual learning, although the paper focuses on the easier problem of task-continual learning where at test time the task id is provided.

The model exploits task similarities to enable the model to transfer knowledge across tasks.
The task similarity is based on the current data of the new task and the data stored in the memory buffer of the old tasks and is computed as the combination of three classical similarity metrics applied on the representation of these data. The paper would benefit from adding a clear motivation of the way this similarity is computed and the intuition behind it.

Now, the overview of the model and the training of the continual learner are described in text. For clarity, this text could be replaced by an algorithm in pseudocode.

The work has value as transfer learning model where you learn to dynamically build your model for a new task exploiting properties of older learned models classified by a number of layer-wise discrete action labels (e.g., reuse, fuse, mask). These action labels are predicted by means of an LSTM that decides which action for which layer. The transfer learning capabilities of the proposed model are clearly shown in the experiments.

The model is evaluated with several datasets commonly used in continual learning and the results of several metrics (accuracy, forgetting metric and transfer ability metric) are convincing that the claims of the paper are accomplished when compared with many baselines.

Remarks to further improve the paper:
-	Line 122: Clarify better why the agent for a task $t$ is trained for $N$ iterations until it converges, where $N$ is the number of tasks.
-	Training time is giving in the supplementary material. An analysis of the computational complexity of the training procedure and ideas for reducing this complexity would strengthen the paper.
-	Figure 1 (A): the font size of the legend could be improved; Figure 1 (B): More explanation in the caption would aid the reader to understand this figure early on when reading the paper.
-	Are the validation sets for reward computation released after publication of the paper?


**Time Spent Reviewing:**

4

---

> ### Author Response · Authors · 2021-08-10
> **Our responses to your comments.**
>
> 1. Re:”Line 122: Clarify better why the agent for a task t is trained for N iterations until it converges, where N is the number of tasks.”
>
> Sorry, this is a double use of the same symbol N. N in line 102 is the number of tasks. N in line 122 is the number of training iterations. The two N’s have different meanings. We will change the N in line 122 to another symbol.
>
> 2. Re:”Training time is giving in the supplementary material. An analysis of the computational complexity of the training procedure and ideas for reducing this complexity would strengthen the paper.”
>
> Usually RL based network structure search takes a relatively long time. One of the reasons for using the action "reuse" to reuse the parameters of some previous tasks is to speed up the convergence of the continual learner, thereby alleviating this problem. As we can see from the steps described in lines 114-121, the complexity is linear in the number of tasks.
>
> 3. Re:”Figure 1 (A): the font size of the legend could be improved; Figure 1 (B): More explanation in the caption would aid the reader to understand this figure early on when reading the paper.”
>
> We will fix these issues in the revised paper.
>
> 4. Re:”Are the validation sets for reward computation released after publication of the paper?”
>         Yes.

---

### Official Review · Reviewer_FW7Z · 2021-07-17

**Rating:** 6
**Confidence:** 3

**Summary:**

The paper proposes an approach for continual learning that builds dynamic network structures such that it minimizes catastrophic forgetting but maximizes forward transfer. The approach involves neural architecture search where an LSTM policy optimizes for selection of network layers (essentially their size) and their initialization with a reward that calculates forward improvement (from the architecture) over iterations as well as backward transfer (across tasks). The results on some standard (albeit toy-ish) CL datasets show that the approach mitigates catastrophic forgetting while also inducing forward transfer in tasks that can use it.

**Limitations And Societal Impact:**

Since the architecture required to solve each task could be expensive (in terms of memory and compute), the authors should potentially discuss this as a limitation.

**Main Review:**

Pros
- The paper is well written. Even though the approach is quite involved, the details are well explained and easy to follow along.
- Experimental results are strong and confirm that the approach can learn architectures for forward transfer while also minimizing forgetting. However, some recent related approaches and baselines are not discussed (see below).
- Ablations are insightful. I also liked the qualitative analysis in the supplementary of a sequence of architectures found by the agent.

Cons
- The datasets used are really small toy datasets, with only a maximum of 10 tasks in a sequence. While these are common datasets used in CL literature, it will be better to move beyond these for more impactful/useful results.
- A recent work [1] seems very closely related. It will be good to compare with this or at the least discuss differences with it. I also see additional baselines in that paper, such as RCL (Xu and Jhu, 2018) and DEN (Yoon et al., 2018) that are also relevant here.
- The approach seems to keep architectures for all previous tasks in memory which would give a prohibitively high memory requirement (order of number of tasks)

Questions for Authors
- The “reuse” action seems to require all parameters of previous tasks in memory. That will have an order of magnitude larger memory footprint than the baselines. Can you comment on this?
- The "new" action seems to double the number of parameters in a layer. This has the potential to blow up the number of parameters in the architecture. How is this managed? Is there a cap on max number of parameters?
- Please clearly characterize the memory requirement, in terms of both the number of parameters as well as the size of replay buffer, that the approach requires.
- Task representations seem to be computed by averaging all the representations of all instances in the task, regardless of their labels (see eq 1). This seems weak and uninformative. Can you comment on what similarities you hope to capture through such representations? Since this is the most important part of the input to the policy, it could be helpful to visualize this similarity for tasks to understand what it is capturing.
- Can you comment on the relevance of [1] and the baselines of RCL and DEN in the context of the proposed approach?

[1] EFFICIENT CONTINUAL LEARNING WITH MODULAR NETWORKS AND TASK-DRIVEN PRIORS, ICLR 2021


**Time Spent Reviewing:**

7

---

> ### Author Response · Authors · 2021-08-10
> **Our responses to your comments.**
>
> 1. Re: ”The datasets used are really small toy datasets, with only a maximum of 10 tasks in a sequence. While these are common datasets used in CL literature, it will be better to move beyond these for more impactful/useful results.”
>
> Actually, we used 50 tasks for CIFAR100 (see Table 2). This large number of tasks is not normally used in existing CL papers. In our future work, we will consider even larger number of tasks and datasets.
>
> 2. Re: “The approach seems to keep architectures for all previous tasks in memory which would give a prohibitively high memory requirement (order of number of tasks)”
>
> The number of parameters of BNS and baselines are given in Table 8 of Appendix A.3. Our current approach saves the model parameters (not the full models) of the past tasks. We plan to try a generative approach in the future to generate the parameters of past models.
>
> About "Questions for Authors"
> 3. Re:”The “reuse” action seems to require all parameters of previous tasks in memory. That will have an order of magnitude larger memory footprint than the baselines. Can you comment on this?”
>
> Existing approaches typically use the network after learning the immediate previous task as the initialization for the new task. The original model parameters for earlier tasks are no longer available as they have been updated/modified by learning of the subsequent tasks. However, as we want to selectively reuse the knowledge learned in each previous task in learning a new task, we save the model parameters of the previous tasks for the purpose. The “reuse” action also enables the continual learner to converge more quickly.
>
> 4. Re:”The "new" action seems to double the number of parameters in a layer. This has the potential to blow up the number of parameters in the architecture. How is this managed? Is there a cap on max number of parameters?”
>
> This may be a misunderstanding. The action “new” is an initialization method, which only initializes the parameters of a layer randomly. It does not change the number of parameter in the layer.
>
> 5. Re:”Please clearly characterize the memory requirement, in terms of both the number of parameters as well as the size of replay buffer, that the approach requires.”
>
> We have given the number of parameters in Table 8 of Appendix A.3. The replay buffer stores a small part of the training data just like other replay-based methods.
>
> 6. Re:”Task representations seem to be computed by averaging all the representations of all instances in the task, regardless of their labels (see eq 1). This seems weak and uninformative. Can you comment on what similarities you hope to capture through such representations? Since this is the most important part of the input to the policy, it could be helpful to visualize this similarity for tasks to understand what it is capturing.”
>
> Our task representation method works well, although it may not be ideal. We actually thought about using the label information. However, if the labels are used, we need to compute and save the representation of each task right after training the task (e.g., by averaging the representations of its data instances in the current continual learner). But as subsequent tasks are learned, the model parameters for each previous task will shift (which causes catastrophic forgetting). Then when a new task arrives, its data representation computed based on the latest model will be in a shifted space and not comparable with the saved representations of the previous tasks in their original spaces when they were learned. The computed similarity will not be reliable. We will explore better similarity methods in the future.
>
> 7. Re:”Can you comment on the relevance of [1] and the baselines of RCL and DEN in the context of the proposed approach?”
>
> MNTDP [1] is not a reinforcement learning (RL) based approach. For each new task, it expands the existing network resulted from the previous tasks by adding modules and selectively connect them to the existing network. DEN [2] is similar to MNTDP (although the details are different) and also not a RL-based approach. Our BNS is very different as BNS is based on RL and it constructs a new network (continual learner) for the new task and initializes it based on 5 actions, not by adding modules to the existing network. RCL (cited in line 76.) is an RL-based method. However, its state is a fixed empty embedding. RCL has only one action, which determines how many filters should be added in each layer for the new task. Its reward is only the new task validation data accuracy and the model complexity. Our BNS uses task similarity as the state of the environment and has 5 actions. BNS’s reward considers both forgetting and transfer and thus can simultaneously tackles both catastrophic forgetting and knowledge transfer. We have experimentally compared with the baseline GPM [3], which outperforms RCL (see the GPM paper [3]). We will discuss in detail the differences between these references in the revised paper.
>
> [1] Tom Veniat, Ludovic Denoyer & Marc’Aurelio Ranzato. Efficient continual learning with modular networks and task-driven priors, ICLR 2021
>
> [2] Jaehong Yoon, Eunho Yang, Jeongtae Lee, and Sung Ju Hwang. Lifelong learning with dynamically expandable networks. ICLR 2018.
>
> [3] G. Saha, I. Garg, and K. Roy. Gradient projection memory for continual learning. ICLR 2021.
>
> 8. Re:”Since the architecture required to solve each task could be expensive (in terms of memory and compute), the authors should potentially discuss this as a limitation.”
>
> Thanks. Yes, we will add this as a limitation. The parameter size and running time are given in Appendix A.3.

---

### Decision · Program_Chairs · 2021-09-27

**Decision:**

Accept (Poster)

**Comment:**

This paper proposes an NAS approach (namely BNS) for continual learning that builds dynamic network structures such that it minimizes catastrophic forgetting but maximizes forward transfer. The experiment results on several CL tasks show the effectiveness of the proposed method.

Overall, all reviewers found the paper to be well written and easy to follow, and focuses on an important research problem with relatively little attention from the area. Based on the reviewer comments, the authors further provided missing details pointed out by the reviewers and some reviewers increased the review score. After the rebuttal, all the reviewers remained positive. I recommend accepting this paper.